# NoEsis: Differentially Private Knowledge Transfer in Modular LLM Adaptation

**Rob Romijnders**,* **Stefanos Laskaridis, Ali Shahin Shamsabadi & Hamed Haddadi**
Brave Research

## Abstract

Large Language Models (LLM) are typically trained on vast amounts of data from various sources. Even when designed modularly (e.g., Mixture-of-Experts), LLMs can leak privacy on their sources. Conversely, training such models in isolation arguably prohibits generalization. To this end, we propose a framework, NoEsis, which builds upon the desired properties of *modularity*, *privacy*, and *knowledge transfer*. NoEsis integrates differential privacy with a hybrid two-staged parameter-efficient fine-tuning that combines domain-specific low-rank adapters, acting as experts, with common prompt tokens, acting as a knowledge-sharing backbone. Results from our evaluation on CODEXGLUE showcase that NoEsis can achieve provable privacy guarantees with tangible knowledge transfer across domains, and empirically show protection against Membership Inference Attacks. Finally, on code completion tasks, NoEsis bridges at least 77% of the accuracy gap between the non-shared and the non-private baseline.

## 1 Introduction

Large Language Models have brought much disruption in the field of Artificial Intelligence and have transformed various use-cases, from intelligent assistants (Dong et al., 2023) and code co-pilots (Chen et al., 2021) to agentic web browsing (Zheng et al., 2024) and enhanced tutoring (Kotalwar et al., 2024). They have shown great scaling potential, devouring terabytes of raw textual or multi-modal data (Kaplan et al., 2020) without their performance plateauing. As this trend continues, all public resources will eventually be consumed. Therefore, tapping into private data silos will become the next significant source of information (Shumailov et al., 2024; Iacob et al., 2024).

However, copyright (Xu et al., 2024) and privacy laws (EU-regulation, 2024; Bukaty, 2019) may prevent models from being trained and served as-is, uniformly across the globe. This introduces the need to orchestrate model training that is somehow separated per region or source. Maintaining separate models, though, quickly becomes intractable and burdensome. Private organizations can own data they want to use for their custom LLM but not expose it publicly Carlini et al. (2021); OpenAI (2023). For instance, client institutions may wish to train domain-specific Copilots (GitHub, 2024) without leaking proprietary information (Niu et al., 2023) to the public domain.

To approach this problem, we draw from Modular Learning (Pfeiffer et al., 2023) for routing knowledge across parts of a neural network and adaptively serve to different domains. While off-the-shelf Mixture-of-Experts (MoE) models (Cai et al., 2024) adopt an architecture where different domains can share common parameters – thus enabling knowledge transfer. However, they can introduce privacy risks (Carlini et al., 2019) exactly because of this sharing. In addition, training an entire MoE model under Differential Privacy (DP) significantly reduces its utility as training a large shared backbone network over multiple domains requires adding large amounts of DP noise. Therefore, there is an inherent tension between *modularity*, *privacy* and *knowledge sharing*.

We propose a privacy-friendly architecture for differentially private modular learning, Nexus-of-Experts (NoE). This inherits a domain-routed mixture of low-rank adapters model (Mix-LoRA), applied on the fully connected layer of each transformer block, further enhanced by shared tokens obtained through a DP training algorithm so that it can be activated across domains (Fig. 1). This architecture allows for a modular architecture and tunably shares knowledge between domains

---

*Work done while interning at Brave Research.

Table 1: Learning approaches and deployment characteristics of each solution. NOEsis is the only method that is *modular*, *private* and can benefit from *knowledge transfer* among domains.

| Modular | Private | Transfer | Approach | Remarks |
|:---:|:---:|:---:|---|---|
| ✗ | ✗ | ✓ | Monolithic model (Wang et al., 2023) | Leaking privacy in model parameters |
| ✗ | ✓ | ✓ | Private learning ((Wang et al., 2023)+(Abadi et al., 2016)) | Low performance |
| ✓ | ✓ | ✗ | MoE (share-nothing) (Zhou et al., 2024) | No benefit from knowledge transfer |
| ✓ | ✗ | ✓ | MoE (shared-expert) (Gong et al., 2022) | Privacy leakage in shared expert |
| ✓ | ✓ | ✓ | NOEsis (ours) | Addresses all three problems |

privately. Compared to conventional MoE methods, we enable efficient document-private learning. Compared to adaptive parameter-efficient fine-tuning (PEFT) alternatives, we facilitate knowledge transfer between domains while providing modularity at deployment time.

In summary, our contributions are the following:

- We expose privacy leakage on existing routing-based models, showcasing the need for privacy-preserving training in these models.
- We propose NOEsis, a domain-routed, hybrid PEFT solution that adopts a shared DP-trained set of prompt tokens that act as a knowledge-sharing backbone and enable knowledge transfer between domains, and a Mix-LoRA that act as private domain experts.
- We analyze the balance between privacy and model expressiveness by tuning the number of trainable parameters and the noise needed to ensure privacy. Indicatively, our hybrid method performs up to 1.4% accuracy points better than the non-hybrid, LoRA-only variant.
- We apply our technique to the problem of code generation, adopting different programming languages as our domains and ensuring privacy at a document level. Our results showcase that NOEsis can achieve from 1.4 to 4.9% points better downstream accuracy, averaged across domains, while protecting our model against privacy leakage.

## 2 DESIDERATA

We study the problem of fine-tuning an LLM for various private domains. This section explains three properties of the problem and shows how NOEsis uniquely satisfies all of them (Table 1).

**Knowledge Transfer.** When training on multiple data sources, knowledge transfer is the improvement in results when learning a joint model from all sources, instead of learning a separate model per source (Ruder et al., 2019). In our multi-domain setting, we desire more accurate predictions in each domain while maintaining privacy regarding potential adversaries from other domains.

**Modular Learning.** Modular learning refers to models whose execution subgraphs can be specialized in terms of function (Pfeiffer et al., 2023; Cai et al., 2024; Fedus et al., 2022). The benefits of modularity for our problem are two-fold. Computationally, modularity allows us to train once and serve model components everywhere without maintaining multiple copies or replicating knowledge. Privacy-related, separating domain-specific and shared parameters allows for a clear separation of concerns in terms of domains. This can be important for serving under different jurisdictions and providing formal guarantees about the flow of information.

**Preserving privacy.** Machine learning models can inadvertently leak sensitive training data (Carlini et al., 2021), which raises privacy concerns when private information is involved. We aim to improve model accuracy by training on multiple private data sets from different domains without impacting their privacy. Therefore, we formalize a threat model when training a model on private datasets from multiple domains, where an adversary in one domain should not be able to discern information about training data in another domain.

These requirements are essential in various real-world scenarios. An example is a globally operating organization that wants to train a single model but serve modularly across different regions and ensure that the model does not leak information between regions due to various copyright or privacy laws (Foo Yun Chee, 2024; Stefano Fratta, 2024). Finally, the modular aspect can be helpful in scenarios where an organization wants to train a model on multiple domains, such as legal, instructional, and administrative documents (Salesforce, 2024), but wants to ensure that the model does not leak information between domains.

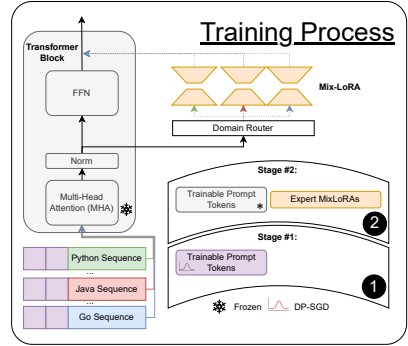
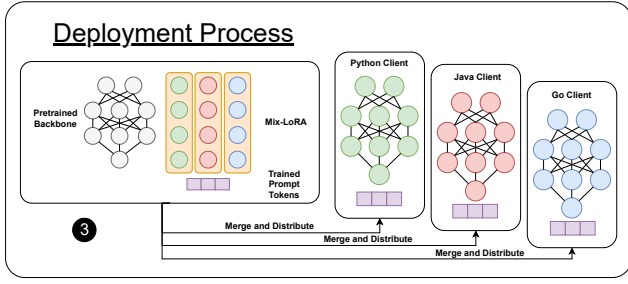

Figure 1: The training and the deployment process of NOEsIS. The training process consists of two stages: Stage 1: The training of private prompt token parameters across domains; Stage 2: The training of expert Mix-LoRA per domain. The deployment involves merging the LoRA parameters with the backbone and sharing the privately trained prompt tokens with each downstream client.

## 3 NOEsIS

Aiming for the properties of *modularity*, *knowledge transfer*, and *privacy*, we introduce NOEsIS, a hybrid, DP-trained, domain-routed Nexus-of-Experts (NoE) solution that adopts a shared set of trainable tokens and Mix-LoRA experts and enables knowledge transfer between domains.

**Domain-routed models.** NOEsIS is fundamentally a modular architecture that enables the specialization of network components to different domains. Given dataset $D_i^{(k)}$ from domain $k$, where $k \in [1, K]$, we route the data from each domain to the respective expert. Instead of operating directly on the backbone model parameters, we use a Mix-LoRA (Hu et al., 2021; Li et al., 2024) architecture that keeps the backbone network frozen and only tunes the adapters per domain. The adapters are applied on top of feed-forward network (FFN) layers of the transformer blocks to enable the "logic" of the network to be tuned per domain Li et al. (2024). Mix-LoRA has been shown to be effective in routing-based neural networks (Li et al., 2024; Feng et al., 2024; Wu et al., 2024; Shen et al., 2024). Formally, let $W \in \mathbb{R}^{p \times q}$ be the weight matrix of a linear layer in the model with parameters $\theta$. We decompose

$$W = W + \alpha \sum_{k=1}^{K} \underline{B^{(k)}} \underline{A^{(k)}}, \tag{1}$$

where $A^{(k)} \in \mathbb{R}^{r \times q}$ and $B^{(k)} \in \mathbb{R}^{p \times r}$ are the trainable (signified by the underline) low-rank matrices for the $k$-th domain-specific adapter, out of $K$ domains. The experimental section refers to the rank $r$ as the *adapter rank*. $\alpha \in \mathbb{R}$ is a scalar constant to reduce the variance impact of the adapter.

**Enabling Knowledge Transfer.** We adopt a differentially private prompt-tuning mechanism to facilitate knowledge transfer between domains while keeping the number of shared trainable parameters minimal. Prompt tuning has emerged as a promising method for efficiently fine-tuning LLMs by introducing differentiable prompts, which are real-valued tokens that can be learned through backpropagation (Li & Liang, 2021; Lester et al., 2021). Prior work demonstrates that DP prompt-tuning can achieve substantial predictive accuracy even with stringent privacy budgets, such as $\varepsilon < 1$ (Duan et al., 2023). By training only the differentiable prompts, we reduce the number of trainable parameters, thereby minimizing the required noise for DP.

Specifically, given input $X = \{x_1^{(k)}, x_2^{(k)}, \ldots, x_L^{(k)}\}$ of $L$ tokens, from domain $k \in K$, a common set of trainable prompt embeddings $P \in \mathbb{R}^{n_{pt} \times d_{\text{vocab}}}$ is prepended to the input sequence, $X' = [P; X]$. $n_{pt}$ is the number of prompt tokens and $d_{\text{vocab}}$ the output dimensionality of the vocabulary. Differentially Private Stochastic Gradient Descent (DP-SGD) (Duan et al., 2023) tunes prompts under the framework of DP by clipping the per-sample gradient to a fixed norm and adding calibrated Gaussian noise to these clipped gradients before applying to update prompts. Concretely, this means $P^{(t+1)} \leftarrow P^{(t)} - \eta \tilde{g}_P^{(t)}$, where $\tilde{g}_P^{(t)}$ is the noisy clipped gradient with respect to the prompt tokens at step $t$ and $\eta$ is the step size.

The **training Process** of NOEsIS is illustrated in Figure 1 (left) and outlined in Algorithm 1 in the Appendix. It follows a two-stage procedure of private learning for the *shared parameters* (Step ❶)

Table 2: The main experimental comparison on modularity, privacy, and knowledge transfer. NOE-SIS uniquely addresses all three aspects, achieving high accuracy while obtaining DP at $\varepsilon = 1.0$ and enabling knowledge transfer across domains. (Knowledge transfer is defined as the increase in accuracy compared to "Share Nothing," which does not have shared parameters between domains.)

| | Model | $\varepsilon$ | Modular | Private | Transfer | PEFT | Python | Java | Go |
|---|---|---|---|---|---|---|---|---|---|
| *(i)* | Share Nothing | 0.0 | ✓ | ✓ | ✗ | ✓ | 68.31 | 60.19 | 64.17 |
| *(ii)* | Solo (separate models) | 1.0 | ✗ | ✓ | ✗ | ✗ | 30.19 | 14.84 | 4.84 |
| *(iii)* | Monolithic Fine-tuning (Abadi et al., 2016) | 1.0 | ✗ | ✓ | ✓ | ✗ | 36.05 | 23.54 | 18.34 |
| *(iv)* | Common LoRA Adapter (rc=512)* (Yu et al., 2021b) | 1.0 | ✗ | ✓ | ✓ | ✓ | 48.21 | 36.16 | 27.24 |
| *(v)* | Prompt-Tuning Only (pt32)† (Duan et al., 2023) | 1.0 | ✗ | ✓ | ✓ | ✓ | 35.03 | 24.29 | 9.67 |
| *(vi)* | NOESIS (pt32)† | 1.0 | ✓ | ✓ | ✓ | ✓ | **69.14** | **61.18** | **66.53** |

*rc, rank $r$ of the common LoRA; †*pt: number of trainable prompt tokens*

and non-private learning for the *domain-specific adapters* (Step ❷). In each iteration of the first stage, a batch of documents is sampled, and the gradients are computed for the domain-specific and shared parameters. The gradients are clipped and noised according to DP-SGD (Abadi et al., 2016). In the second stage, SGD refers to Stochastic Gradient Descent. During deployment (Step ❸), the LoRA parameters are sent to each respective domain and merged with the backbone.

For **document-level DP,** we use the definition from Dwork & Roth (2014). A randomized algorithm $f(\cdot)$ is $(\varepsilon, \delta)$-differentially private if the following holds for any two adjacent data sets $D, D'$, and for any subset $\mathcal{S}$ of outputs: $\Pr[f(D) \in \mathcal{S}] \leq e^{\varepsilon} \Pr[f(D') \in \mathcal{S}] + \delta$. Here, $D = \{d_i\}_{i=1}^{N}$ is a dataset of $N$ documents. Each $d_i$ is a string of tokens of arbitrary length. In the multi-domain setting, a dataset comprises all documents in all domains. Two datasets $D$ and $D'$ are adjacent when they are identical except for one document in any domain being removed. For this privacy protection, we can calculate the noise multiplier as a function of total dataset size $N$ and batch size $N_b$ in Algorithm 1. In keeping with prior work, we run all privacy-focused experiments with $\varepsilon = 1.0$ and $\delta = 10^{-6}$, which is smaller than the rule-of-thumb of one divided by dataset size (Hsu et al., 2014).

## 4   EVALUATION

Our evaluation adopts the task of code completion among different programming languages, which we consider private domains. First, we evaluate the knowledge transfer between domains. Second, we perform a sensitivity analysis for the number of trainable parameters. Finally, we run a privacy attack to investigate privacy leakage empirically.

**Model.** We use a decoder-only model architecture for our experiments (Radford, 2018) – specifically, the pre-trained CodeT5+ (220M) model. Given the previous tokens as input, the model is trained to autoregressively predict the next token in a sequence by using the softmax as likelihood and cross-entropy as the loss function. This setup is particularly suitable for programming language tasks, where the goal is to generate code one token at a time (Gong et al., 2022).

**Datasets.** We use the Python, Java, and Go programming languages for our multi-domain training because the pre-trained model was trained on data specifically deduplicated for datasets in these languages (Wang et al., 2023). The deduplication is necessary to simulate private domains for fine-tuning. The dataset for Python is the PY150 dataset (Raychev et al., 2016) and for Java the JavaCorpus (Allamanis & Sutton, 2013), which follows a similar setting as Gong et al. (2022). The Go data comes from the CodeSearchNet dataset (Husain et al., 2019). More details about the datasets can be found in Appendix C.1.

**Training Setup & Hyperparameters.** We train our models on 4×4090 GPUs, using a learning rate of $10^{-3}$. We use the privacy-accountant of the Opacus library to calculate the noise-multiplier constant (Yousefpour et al., 2021). An epoch is defined as sampling one 512-token block for each document in the dataset. This ensures optimal use of compute resources and maintains the privacy guarantee as each epoch uses a document exactly once. Details about our hyperparameter values can be found in Appendix C.3. The code to run the experiments corresponding to Table 2 can be found at the anonymized GitHub: www.anonymous.4open.science/r/noesis-48E5.

We compare NOESIS against various trainable baselines, namely: *i) Share Nothing*: a modular model without shared parameters, which is to simulate the effect of not having knowledge transfer; *ii) Solo*: a separate model fine-tuned separately for each domain; *iii) Monolithic Fine-tuning*: DP fine-tuning the whole model without any routable components (Abadi et al., 2016), which is to compare with Solo and confirm the effect of knowledge transfer in monolithic models (Hokamp

Table 3: Prompt-tuning achieves the best trade-off between the number of shared parameters and accuracy, across domains. All results obtained under the privacy guarantee.

| Model | # Shared Params. | Python | Java | Go |
|-------|------------------|--------|------|-----|
| rc64 | $7680 \times 768 = 5.9M$ | 68.73 | 60.32 | 65.32 |
| rc4 | $480 \times 768 = 369k$ | 68.77 | 60.45 | 65.39 |
| rc1 | $120 \times 768 = 92k$ | 68.74 | 60.43 | 65.26 |
| pt120 | $120 \times 768 = 92k$ | 69.13 | 61.09 | 66.49 |
| pt32 | $32 \times 768 = 25k$ | **69.14** | **61.18** | **66.53** |
| pt8 | $8 \times 768 = 6k$ | 69.13 | 61.16 | 66.49 |

rc, rank $r$ of the common LoRA; pt, number of prompt tokens

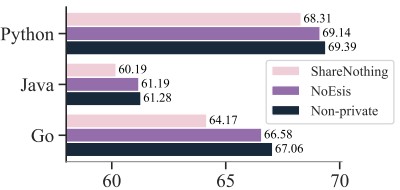

Figure 2: Between the results of a non-shared model, which is the baseline, and a non-private model, which obtains the highest accuracy, NOESIS bridges the accuracy gap by more than 77%.

et al., 2019); *iv) Single Common Adapter*: A common LoRA trained with DP across domains (Yu et al., 2021b), which is to compare the influence of using PEFT instead of monolithic fine-tuning; *v) Prompt-Tuning Only*: tuning only the prompts under the DP guarantee, which is to investigate the impact of knowledge transfer without any modularity in the form of Mix-LoRA (Duan et al., 2023). Details about the baselines can be found in Appendix C.2.

**Experimental Results.** The results of comparing NOESIS against baselines are shown in Table 2. Compared to Share Nothing, we see that NOESIS achieves an improvement of 0.83% points on Python, 1.0% points on Java, and 2.41% points on Go. This indicates that knowledge transfer is beneficial for next token prediction, especially in the case of a scarce domain, such as Go, which has $50\times$ less data than Python. Appendix D.4 shows qualitative examples of this comparison. To aid interpretation of numerical results, Appendix Figure 7 shows the results of NOESIS run on five different random seeds, making clear that generally the standard deviation is around 0.02% point.

Comparing NOESIS to Prompt-Tuning Only, we observe an improvement of more than 30% points across the three domains. This shows the benefit of using modularity as a privacy-friendly architecture. In general, the modular models achieve higher accuracy than the non-modular models, indicating the importance of modularity in multi-domain training. Comparing the experiments of Solo and Monolithic, we observe the effect of positive knowledge transfer in the non-modular setting, as the latter model achieves higher accuracy than each of the solo models. Finally, using a Common LoRA Adapter achieves about nine accuracy points improvement compared to monolithic fine-tuning, which corroborates earlier studies on PEFT and DP (Yu et al., 2021a;b).

Investigating the effect of knowledge transfer, we ask *"how many shared parameters are optimal for knowledge transfer?"* This is contended by three directions: having too many parameters is vulnerable to *overfitting*; having too few can lead to *underfitting*; finally, more shared parameters require more noise in algorithm that satisfy DP (Abadi et al., 2016), which negatively influences learning. To investigate this contention, we train with $4\times$ more and $4\times$ less trainable prompt tokens. We also evaluate a variant with a LoRA as the trainable parameters for knowledge-sharing (explained in Appendix C.2.1). From Table 3, the model with 32 shared prompt tokens (pt32) achieves the highest accuracy across all domains. This indicates that a moderate number of shared parameters appears optimal for knowledge transfer. Using a LoRA as a common parameter generally has lower accuracy than any prompt-tuning method. *This indicates that prompt-tuning is a parameter-efficient option for providing knowledge transfer under differential privacy.*

Finally, we quantify the "cost of privacy" by comparing NOESIS with the same setup without the DP guarantee. As shown in Fig. 2, between the Share Nothing baseline (non-shared) and the non-private variant, NOESIS is able to achieve 84% of the way on average. *This indicates that NOESIS effectively achieves knowledge transfer and high accuracy while guaranteeing privacy.*

**Empirical Assessment of Privacy.** Complementing the analytical privacy guarantees, we measure empirical privacy based on a Membership Inference Attack (MIA) (Shokri et al., 2017). The attack aims to determine whether an input sequence, $s$, is a member of the model's training data. Several MIAs on LLMs have been proposed (Chang et al., 2024; Carlini et al., 2021), and we leverage an MIA based on the predictive log-likelihood scores and extend the attack to a multi-domain setting, named *cross-domain MIA*. The attack is particularly relevant for modular models (Lepikhin et al., 2020; Gong et al., 2022; Gururangan et al., 2022; Zhou et al., 2024; Komatsuzaki et al., 2022; Wu et al., 2024), where shared parameters can leak information between domains.

The cross-domain attack follows a threat model where an attacker can make predictions using the parameters in a particular domain and aims to infer the membership of training data in another domain. The scoring function $S(\mathbf{s}) = \mathcal{L}(\mathbf{s}; \mathcal{M})$ is the average log-likelihood assigned to each successive token. The attacker compares this to a threshold $\tau$ to predict the membership. The attack success is measured with Area Under the ROC Curve (AUC) and the True Positive Rate (TPR) – defined in Appendix C.4. The results in Table 4 demonstrate that the original Mix-LoRA models can leak private information. For example, attacking Java via the Go domain, NoEsis has less empirical privacy leakage, measured by AUC, from 65.4 to 55.7% for

Table 4: Mix-LoRA models have a privacy vulnerability in parameters that are shared between domains. NoEsis reduces this vulnerability while maintaining good predictive accuracy (Figure 2). The results are formatted as 'non-private result' → 'private result'.

| Attack | AUC (%) | TPR@1 (%) |
|---|---|---|
| Python (via Java) | $51.9 \rightarrow 50.6$ | $1.1 \rightarrow 1.0$ |
| Python (via Go) | $51.7 \rightarrow 50.7$ | $1.2 \rightarrow 1.1$ |
| Java (via Go) | $65.4 \rightarrow 55.7$ | $2.6 \rightarrow 1.4$ |
| Java (via Python) | $64.8 \rightarrow 55.0$ | $2.3 \rightarrow 1.3$ |
| Go (via Python) | $53.2 \rightarrow 50.6$ | $1.1 \rightarrow 0.7$ |
| Go (via Java) | $54.0 \rightarrow 50.1$ | $1.3 \rightarrow 0.6$ |

NoEsis. Following Carlini et al. (2022), Table 4 reports the TPR at a 1% False Positive Rate (FPR), which indicates attack success when the attacker makes only 1% false positives. Across all domains, the attack on NoEsis has a lower (better) TPR compared to a Mix-LoRA model.

## 5 RELATED WORK

**Routing-based models** have been extensively studied to build modular language models (Cai et al., 2024; Fedus et al., 2022). MoE architectures consist of "expert" sub-models, each specialized for a particular domain or task, and a "router" that selects which expert to activate for a given input sequence or token. Multiple works have combined modular models and PEFT (Ding et al., 2023; Hu et al., 2021). Li et al. (2024) introduce Mix-LoRA for memory efficiency in language modeling; Wu et al. (2024) introduce Mix-LoRA to study domain specialization; and Shen et al. (2024) introduce Mix-LoRA for multi-modal instruction tuning. Most similar to ours, Feng et al. (2024) introduces Mix-LoRA for routing on domains like finance, medicine, and coding. Other PEFT approaches include prefix-tuning (Li & Liang, 2021) or adapter learning (Houlsby et al., 2019; He et al., 2021). We choose a hybrid of prompt-tuning (Lester et al., 2021), which has been shown effective for privacy-aware fine-tuning (Duan et al., 2023), and Mix-LoRA for effective domain experts.

**Preserving privacy** has mostly been studied with DP (Dwork & Roth, 2014). However, DP approaches have primarily focused on applying DP to all model parameters (Yu et al., 2021a; Li et al., 2021). The benefits of leveraging public data during private fine-tuning were studied (Kerrigan et al., 2020; Golatkar et al., 2022), showing that access to public data can improve the performance of DP-trained models. In our case, we use public data as the pre-trained model initialization and fine-tune with private datasets. Similar to our setting, Tholoniat et al. (2024) studies DP in an MoE setting but focuses on learned routers and emergent expertise, whereas we aim at private learning with positive transfer between domains. Regarding the MIA, Hu et al. (2022) explores a multi-modal attack, but our is about the multi-domain setting with access to predictive log-likelihoods.

**Knowledge Transfer in Modular Models** has been studied with large-scale pre-training with various data sources (Radford, 2018; Devlin et al., 2019; Touvron et al., 2023). The term was introduced by Hokamp et al. (2019) and has been observed in the context of modular language models in the MultiCoder model (Gong et al., 2022). However, that work does not consider the privacy of each domain, nor does it study the privacy leakage in shared parameters of a neural network.

## 6 CONCLUSION

In this work, we have introduced a first-of-its-kind, hybrid PEFT architecture that we call Nexus-of-Experts. Our method, NoEsis, establishes a framework for modular learning that enables knowledge transfer across domains that contain private documents. We apply our technique to the problem of multi-lingual code completion and our evaluation demonstrates that NoEsis effectively balances privacy and utility, achieving high accuracy across domains while providing strong guarantees at $(\varepsilon = 1., \delta = 10^{-6})$-DP. NoEsis bridges the accuracy gap between non-private and non-shared models by over 77%. At the same time, we uncover an empirical privacy leak in the baseline modular models and show how our method can defend against such attacks and protect training sources.

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

## A    LIMITATIONS & FUTURE WORK

We have laid the foundations for a new, privacy-friendly, modular method of addressing domain privacy in LLMs. However, this is an important problem with many additional dimensions to consider.

The approach has been tested against a relatively small LLM, i.e., CodeT5+ (220M parameters) and a single dataset (CODEXGLUE). While enough for making a case towards this new paradigm, we aim to extend our evaluation to additional domains and tasks, such as use-cases in Natural Language. Moreover, while largely orthogonal, we have leveraged LoRAs as our PEFT technique of choice for domain experts. Nevertheless, there have been various alternative architectures for adaptation, including but not limited to DoRA (Liu et al., 2024) and VeRA (Kopiczko et al., 2024). We aim to explore such alternative forms for parameter-efficient adapters and their interplay with NoE in future work. There is also the question of achieving modularity, privacy, and knowledge transfer in a single-stage training method, instead of the two stages in Algorithm 1, but that must ensure that gradients from private and locally non-private samples do not interact. Finally, we uncover and defend against a single type of attack, namely the Membership Inference Attack. However, there are different possible threat models and attacks (e.g., adversarial (Zou et al., 2023) and data reconstruction (Carlini et al., 2019)). While not in the direct scope of our study, we do acknowledge their importance in various deployment settings and future avenues of work.

## B    ABLATION STUDY

Table 5: Ablation: Surgically removing the shared parameters, after the training has finished, leads to the most accuracy decrease in the domain with the most data, Python. Removing the domain experts leads to the most accuracy decrease in the domain with the least data, Go.

| Model | $\varepsilon$ | Python | Java | Go |
|---|---|---|---|---|
| NOESIS (pt32) | 1.0 | 69.14 | 61.19 | 66.58 |
| w/out common prompts | 0.0 | 4.65 | 14.31 | 15.42 |
| w/out domain experts | 1.0 | 33.91 | 24.30 | 19.13 |

We perform an ablation study to understand the impact of each component of NOESIS on the final performance. In this setting, we surgically remove the shared parameters to investigate the impact on the results. The results are shown in Table 5. First off, we observe that removing any of the components of NOESIS is detrimental to its downstream accuracy, with a minimum drop of 36.89% points and an average of 50% for common prompts and 30% for domain experts. The common prompt seems to be more important in the downstream accuracy, but this can be an effect of our training stage ordering. However, it yields an important conclusion that LoRA experts do not learn or "undo" the knowledge of prompt tokens, but incorporate it into their knowledge. Moreover, the common prompts are not a nuisance and do encode important bits of information across domains. Finally, and in line with the comparison with the Share-Nothing (*(i)*), it seems that the positive transfer (whether by removal of common prompts or by fine-tuning without them) more severely affects smaller datasets and especially Go with $50\times$ fewer documents than Python. Therefore, scarce datasets tend to benefit more from knowledge sharing.

For the analysis in Fig. 2, the results for the scarce language, Go, are particularly noteworthy. The knowledge transfer is more than 2.4% points over the Shared Nothing setup. *This highlights the effectiveness of our approach in leveraging shared knowledge across domains to enhance token completion tasks, especially for a lower-resource domain.*

## C    EVALUATION SETUP DETAILS

### C.1    DATASET DETAILS

This section provides additional details about the datasets we use for training, their partitioning, and deduplication against the pre-training set of CodeSearchNet.

**Python Dataset.**    The PY150 data set consists of 150,000 Python repositories (Raychev et al., 2016). Each "document" is the code from a GitHub repository collected in 2016. The original pub-

Table 6: An overview of datasets and domains used for the training and evaluation of NOESIS on code completion tasks.

| Domain | Source | Description | Training Set | Evaluation Set |
|---|---|---|---|---|
| Pre-training | GitHub Code (CodeParrot, 2025) | Upstream data | 3.7 million | |
| Python | PY150 (Raychev et al., 2016) | Published dataset[†] | 100,000 | 50,000 |
| Java | Java GitHub Corpus (Allamanis & Sutton, 2013) | Published dataset[†] | 12,934 | 8,268 |
| Go | CodeSearchNet (Husain et al., 2019) | Extracted from CodeSearchNet [†] | 2,000 | 1,559 |

[†] Dataset explicitly deduplicated in pre-training dataset (Wang et al., 2023).

lishers of the data set did select for repositories with permissive licenses. While originally created for code parsing, we use only the actual code tokens for next-token prediction.

**Java Dataset.** The JavaCorpus data set consists of $14,807$ Java code repositories (Allamanis & Sutton, 2013). Each "document" is the code from a GitHub repository collected in 2013. The original publishers of the data set did select for repositories with permissive licences and repositories that were forked at least once. Additionally, repositories with a "below average" reputation score on GitHub are not considered. In 2021, the CODEXGLUE effort repeated this strategy to collect another $6,395$ code repositories under the same collection policy (Lu et al., 2021).

**Go dataset.** For Go, we make a new train-test split from the CodeSearchNet dataset, which we know was also deduplicated against by (Wang et al., 2023). As CodeSearchNet has data for a text-to-code search task, we remove the search queries and concatenate all functions of a repository to obtain a document. The data set sizes are noted in Table 6.

The data set for Go consists of $167,288$ search queries originating from $3,559$ repositories. We split the data set into $2,000$ repositories for training and $1,559$ repositories for testing. Functions are concatenated with interleaved linebreaks. This mimics the provisioning strategy for the PY150 and JavaCorpus datasets, where each document represents a complete code library and thus contains multiple functions.

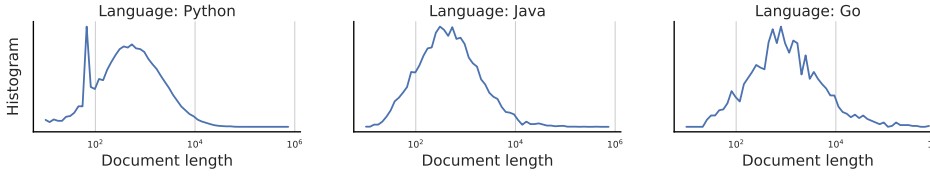

Figure 3: Histograms of number of tokens per document for the three domains. The distributions are similar in order of magnitude. The dataset sizes are different: Python has $100,000$ documents for training, Java $12,934$, and Go $2,000$.

**Histograms:** Fig. 3 shows histograms of the number of tokens per document for the training sets of the three domains. This indicates that the described provisioning method for Go has a similar order of magnitude of tokens per document as the Python and Java datasets. The histogram for Go is more noisy since the dataset contains only $2,000$ documents, compared to $100,000$ documents for Python. The histogram for Python shows a single peak at 81 tokens per document, which is a boilerplate script that is repeated several times in the training set.

We use the data sets as they are tokenized by CODEXGLUE. Long tokens might be split up by the CodeT5+ tokenizer, similar to the training algorithm of the pre-trained model (Wang et al., 2023). Examples of tokens as blank-spaced strings are in Section D.4.

## C.2 BASELINE DETAILS

In this section, we describe each baseline in greater detail and why we compare to it. Specifically:

*i)* **Share Nothing** (Fig. 4a): This setup represents the architecture where we train domain adapters of a model without any common parameters. The backbone model remains frozen. This effectively

represents a setup which can have no knowledge transfer between the individual domains. In other words, there is no parameter whose gradient depends on more than one domain.

*ii)* **Solo** (Fig. 4b): Solo represents a set of models, each trained with DP-SGD individually on a specific domain. In this case, the baseline only shares the same pre-training between the individual models and diverges during private fine-tuning.

*iii)* **Monolithic Fine-tuning** (Fig. 4c): This setup represents the full fine-tuning of a single model on all datasets jointly. It exemplifies the paradigm with the largest potential for positive transfer but is also prone to overfitting as the setup has many trainable parameters.

*iv)* **Single Common Adapter** (Fig. 4d): This setup represents the LoRA variant of *(iii)*, where we tune a common LoRA with rank four across all domains. This baseline is to corroborate the increase in accuracy when using PEFT in a training algorithm with DP guarantees (Yu et al., 2021a;b).

*v)* **Prompt-Tuning Only**(Fig. 4e): This baseline represents the training of 32 common prompt tokens across all domains only, with the rest of the backbone being frozen. It represents the most lightweight PEFT variant here.

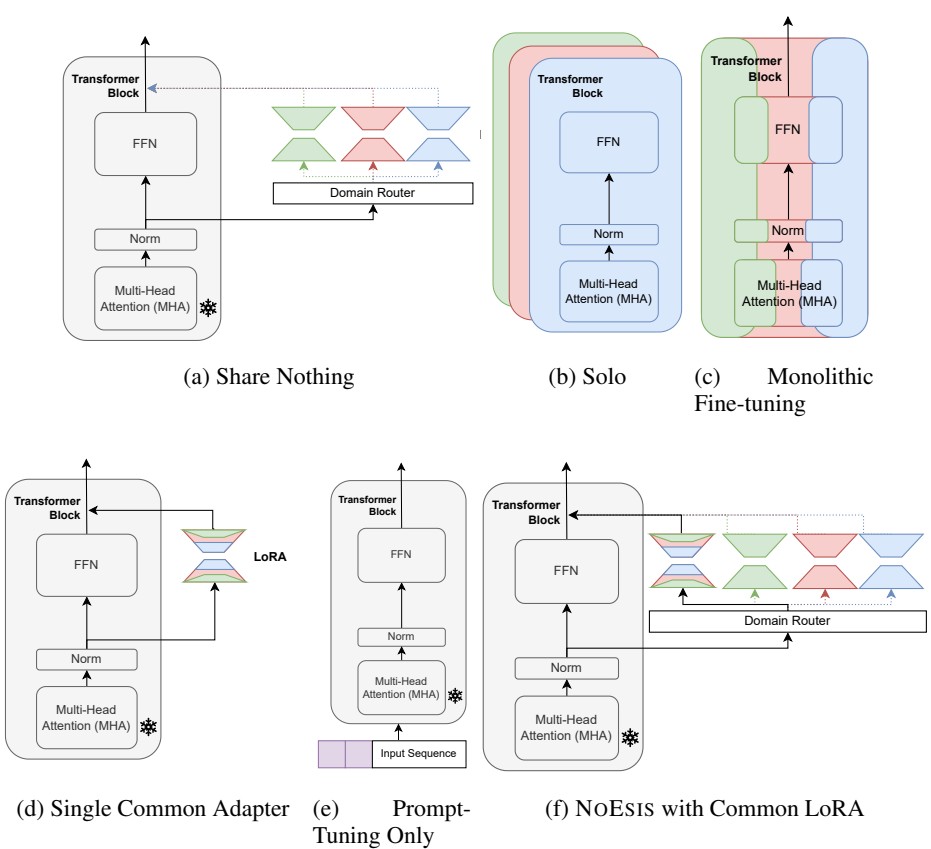

(a) Share Nothing      (b) Solo      (c)    Monolithic Fine-tuning

(d) Single Common Adapter    (e)      Prompt-Tuning Only      (f) NOESIS with Common LoRA

Figure 4: Illustration of different baselines. Grey represents frozen, while blue, green, and red represent the domains with which we train that specific component. Purple represents trainable prompts.

### C.2.1 NOESIS WITH LORA AS KNOWLEDGE SHARING BACKBONE

Table 3 compares NOESIS against a variant with the "shared parameters" in the form of another LoRA (Feng et al., 2024). We visually present this method in Fig. 4f.

Concretely, this corresponds to a modification of Equation 1:

$$W = W + \alpha \left( \underline{B^{(c)} A^{(c)}} + \sum_{k=1}^{K} \underline{B^{(k)} A^{(k)}} \right). \tag{2}$$

Here, $A^{(k)}$ and $B^{(k)}$ are the trainable parameters as explained before. $A^{(c)} \in \mathbb{R}^{r_c \times q}$ and $B^{(c)} \in \mathbb{R}^{p \times r_c}$ constitute the LoRA decomposition, with $r_c$ corresponding to the rank of the common adapter, which is mentioned in the first column of Table 3. All results in this table are obtained under $(\varepsilon = 1.0, \delta = 10^{-6})$-DP.

For the domain-specific parameters, in CodeT5+ , there is a bottleneck structure in $W$ of $W_i^{d_{\text{model}} \times d_{\text{ff}}}$ and $W_o^{d_{\text{ff}} \times d_{\text{model}}}$, where $d_{\text{model}}$ is the model hidden size ($d_{\text{model}} = 768$, $d_{\text{ff}} = 2048$ for CodeT5+ ) and we have one LoRA for each matrix.

### C.3 NoEsis Hyperparameter Details

**Batch Size.** Keeping the ratio of compute to memory optimal, we run all experiments with batch size $N_b = 64$ on nodes with $4\times$ 24gb-RAM GPUs. The first stage of Algorithm 1 is trained with batch size 96, as more memory is available when the domain experts are not yet used. One notable exception is the pt120 experiment in Table 3, which has only batch size 64 due to the large memory requirements of (gradients and activations for) the prompt tokens.

Table 7: Hyperparameter overview of the learning rate used in the second stage of Algorithm 1.

| Learning Rate | Python | Java | Go |
|---|---|---|---|
| $10^{-4}$ | 67.65 | 61.02 | 64.90 |
| $10^{-3}$ | 69.14 | 61.19 | 66.58 |
| $5 \cdot 10^{-3}$ | 51.17 | 46.34 | 28.96 |

**Optimizer & Learning Rates.** For all training algorithms, we use the AdamW optimizer (Loshchilov & Hutter, 2019) with a learning rate $10^{-3}$, similar to prior work (Wang et al., 2023). A small sweep of learning rates in Table 7 shows that either a smaller or larger learning rate results in lower accuracy. For scheduling, we use a linear step scheduler with 500 steps of warmup.

**Rank Size and # Trainable Tokens.** When performing parameter-efficient fine-tuning with LoRA, the rank we select depends on whether the respective component is trained with DP or not. For the domain adapters, we use domain experts of $r = 512$, whereas for the case of common LoRA, we use $r = 4$ (as shown also in Tab. 3). We select rank 4 of the common adapter using Table 3, and in earlier research phases we used a finer sweep. For selecting the rank 512 for the domain adapters, we find that a higher rank generally has better results, however, beyond rank 512, the model would take up too much GPU memory.

**DP Epsilon and Gradient Clipping Norm.** Throughout our experiments in the evaluation, we report results for $\epsilon = 1.0$. However, we have experimented with epsilon values of $\{0.01, 0.1, 1.0, 8.0, 16.0\}$. We report the best results that guarantees privacy without significant utility tradeoff (i.e., more than one percentage point of accuracy degradation).

For gradient clipping, we have experimented with values in the range of $[0.01, 10]$ in exponential steps and have used 1.0 as the value of choice. Using a value of 0.1, for example, yields a slightly worse performance of $69.13, 61.16, 66.48$ on Python, Java, and Go, respectively, for a NoEsis pt32 model. Similarly, the rc4 model in Table 3 would decrease in accuracy to $68.75, 60.32, 65.32$.

**Training settings.** Our method is built with PyTorch (v2.4.1) and HuggingFace transformers (v4.45.0), with the functionality for DP handled by Opacus (v1.5.2). We train our models on $4\times4090$ GPUs. During training, we use a context length of 512 tokens and a learning rate of $10^{-3}$. For training under the DP guarantee, we apply gradient clipping with a norm of 1.0 and add Gaussian noise with a standard deviation, which is scaled according to the targeted privacy budget. We use the privacy-accountant of the Opacus library to calculate the noise-multiplier constant (Yousefpour et al., 2021). A private document within the dataset can be of any length. Therefore, an epoch is defined as sampling one 512-token block for each document in the dataset. This ensures optimal use of compute resources and maintains the privacy guarantee as each epoch uses a document exactly once.

## C.4 DETAILS ON THE MEMBERSHIP INFERENCE ATTACK

In this section, we provide additional details on the implementation and results of the Membership Inference Attack. The attack and privacy protection goes with the following use case: $K$ institutions hold data for their own domain. For example, one institution holds Python data, another holds Java data, and a third holds Go data. A trusted server receives all the data to train with cross-domain insights, e.g., knowledge transfer. The trusted server returns to each institution the domain-specific model parts and the shared parameters, which are then used for inference. The privacy of each domain is maintained, and the model can simultaneously benefit from the shared knowledge. The process is illustrated in Fig. 1 (right).

For the results, the definition of TPR and FPR are particularly relevant:

- TPR is the proportion of members of the training set that are identified as members, i.e., for which $S(\mathbf{s}) > \tau$.
- FPR is the proportion of non-members that are incorrectly identified as members, i.e., for which $S(\mathbf{s}) > \tau$.

From these rates, we report two aggregate metrics, both of which are common in the literature (Carlini et al., 2022):

- The **AUC** is a measure of the trade-off between TPR and FPR. A value of 1.0 indicates the worst possible empirical privacy protection, while 0.5 indicates good empirical privacy protection. Fig. 5 shows the ROC curve for the cross-domain MIA per domain. The AUC is calculated by plotting the TPR against the FPR for all possible threshold settings and measuring the area under this curve. This corresponds to the METRICS.AUC method in Pedregosa et al. (2011).
- The **TPR @1** reflects the attack accuracy rate when making a minimal number of false positives, i.e., 1% FPR. For this evaluation, we find the threshold $\tau$ where the FPR is 1% and report the TPR. If such a threshold does not exist, we take a weighted average of the TPR at the nearest two $\tau$ with lower and higher FPR.

The score function $S(\mathbf{s})$ is defined as the average log-likelihood assigned to each successive token of the input text by the model. This is also named *teacher forcing* in the literature. In practice, this is implemented in a sequence of 512 tokens, like the training setting:

$$S(\mathbf{s}) = \mathcal{L}(\mathbf{s}; \mathcal{M}) = \frac{1}{L-1} \sum_{l=2}^{L} \log \left( \mathcal{M}(s_l | s_{1:l-1}; D) \right) \tag{3}$$

where $L$ is the sequence length, 512, $\mathcal{M}$ is the autoregressive model assigning log-likelihood for token $s_l$, conditioned on tokens up to $l-1$, $s_{1:l-1}$ and having been trained on dataset $D$. The threshold $\tau$ is varied across all possible scores when evaluating the ROC curve. The input tokens $s_{1:l-1}$ are from the training and test set as indicated in Table 6. The studied Mix-LoRA model is trained with SGD, like the non-private result in Fig. 2.

Extending the results of Fig. 5, we provide the ROC-AUC curves for all the combinations of Python, Java, and Go source and target languages in Fig. 5 below. We witness that for NOESIS, the TPR is closer to the random chance line, which is when the TPR equals the the FPR.

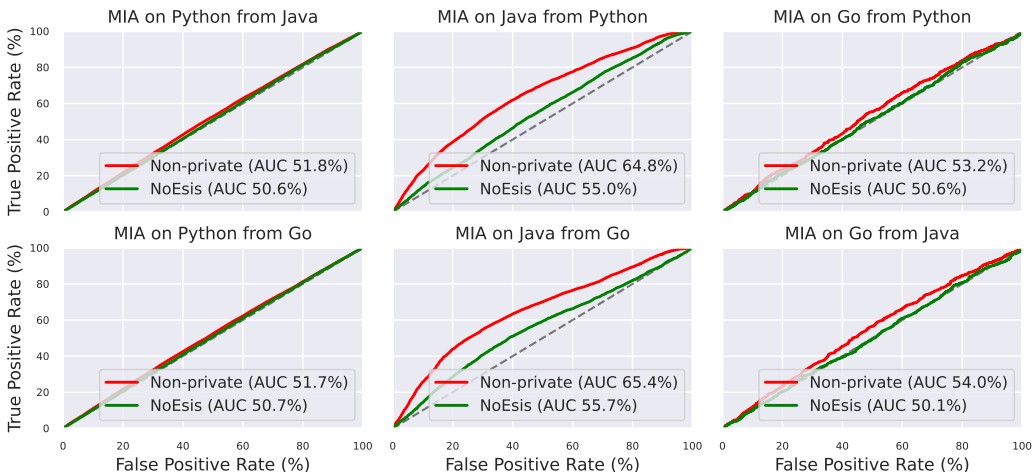

Figure 5: Visualization of results in Table 5 with all combinations of source and target domains.

## D    NOESIS OPERATIONAL DETAILS

### D.1    ALGORITHM

The algorithm for training NOESIS is outlined in Algorithm 1. It follows a two-stage procedure of private learning for the *shared parameters* (Step ❶) and then non-private learning for the *domain-specific adapters* (Step ❷). The parameters for the expert LoRA are collectively referred to as $E$ and comprise all $B$ and $A$ matrices in Equation 1: $E = \{A^{(k)}, B^{(k)}\}_{k=1}^{K}$. The private parameters are referred to as $P$, defined in Section 3. The algorithm starts by computing the noise multiplier $\sigma$ using the privacy accounting mechanism (Yousefpour et al., 2021). The dataset is augmented with domain labels, which are used for deterministic routing. In each iteration, a batch of documents is randomly drawn from all domains, and the gradients are computed for the domain-specific and shared parameters. In the case of private learning, the gradients are clipped and noised according to DP-SGD (Abadi et al., 2016). In the second stage, SGD refers to Stochastic Gradient Descent.

---

**Algorithm 1** Stepwise Training of NOESIS

**Input:**    Pre-trained model $\theta_0$, training sets $\{D_k\}_{k=1}^{K}$ for K domains, collectively named $D$ with $N$ documents, batch size $N_b$, learning rate $\eta$, number of iterations $T_1$, $T_2$, privacy budget $\varepsilon, \delta$, gradient clipping bound $C$
**Output:** Modular LLM with privacy guarantees

*Stage 0: Compute noise multiplier*
$\sigma \leftarrow$ PRIVACCOUNTING$(\varepsilon, \delta, B, N, T_1)$
*Stage 1: DP training of shared parameters*
$P_0 \leftarrow$ Randomly initialize prompt tokens
**for** $t = 1$ **to** $T_1$ **do**
    Randomly draw batch $\mathcal{B}_t$ of size $N_b$ from $D$
    $P_t \leftarrow P_{t-1} -$ DP-SGD$(\theta_0, P_{t-1}, \mathcal{B}_t, \sigma, C)$
**end for**
*Stage 2:  Training of domain-specific parameters*
$E_0 \leftarrow$ Randomly initialize expert adapters
**for** $t = 1$ **to** $T_2$ **do**
    Randomly draw batch $\mathcal{B}_t$ of size $N_b$ from $D$
    $E_t \leftarrow E_{t-1} -$ SGD$(\theta_0, P_T, E_{t-1}, \mathcal{B}_t)$
**end for**

---

### D.2    LEARNING CURVES

To assess the convergence of our training runs, Fig. 6 displays the training loss and gradient norm during the training of the models. As expected, both the loss and gradient norm decrease during the twelve training epochs. During the first stage, shown in the top row, the empirical gradient norm before clipping takes values around 1, which also happens to be our clipping norm. We have experimented with a lower clipping norm but found that this would result in lower accuracy. While the training losses for the second stage, bottom row, seem similar, training 32 prompt tokens achieves the highest accuracy on the test set, see Table 3. The gradient norms are calculated for the shared parameters $P_t$ (Algorithm 1) in the top row and for expert parameters $E_t$ in the bottom row. The

training loss is the cross-entropy log-likelihood averaged along the sequence and averaged among all three domains.

## D.3 RANDOMNESS IN DP TRAINING

To explore the stochasticity introduced by training with a differentially private algorithm, we retrain NOESIS using five different random seeds. By running multiple training sessions with other seeds for the additive noise, we can better understand the range of possible outcomes. The results of these experiments are summarized by the boxplots in Figure 7. The box edges indicate the lower and upper quartile and the whiskers correspond to the minimum and maximum values among the five random seeds.

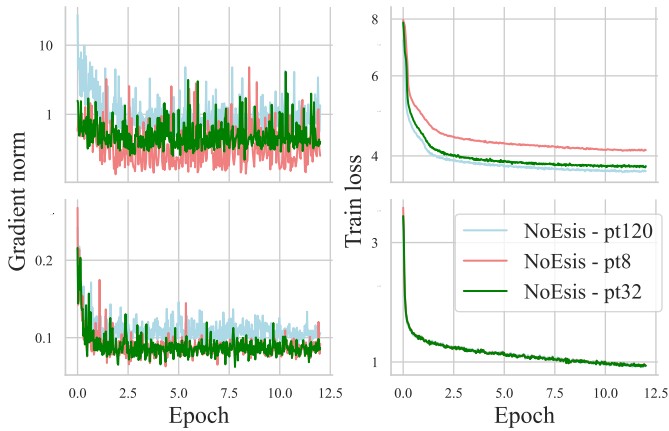

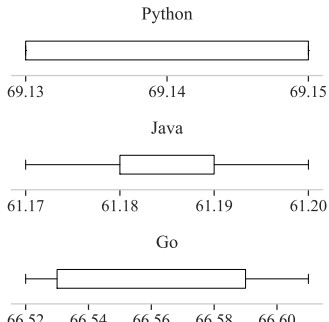

Figure 6: The loss and gradient norm during training for the ablation experiment in Table 5. The bottom row illustrates the second training stage. While the training losses appear similar, training with 32 prompt tokens achieves the highest test accuracy.

Figure 7: Illustrating the stochasticity due to training NOESIS with DP-SGD. Python, having the most data, shows the least variability, while Go, as the scarce domain, exhibits greater variability in accuracy.

## D.4 QUALITATIVE EXAMPLES OF MISPREDICTIONS

In this analysis section, we provide qualitative examples of mispredictions made by the model. These examples indicate the differences in performance between the ShareNothing model and NOESIS and highlight the benefit of knowledge transfer. We use color coding to demonstrate the correctness of the predictions: **red** indicates that both models predict the token incorrectly, **blue** indicates that ShareNothing is correct but NOESIS is not, and **green** indicates that ShareNothing is incorrect but NOESIS gives the correct prediction. The datasets in CODEXGLUE are provided stripped and tokenized, so syntax highlighting and indentation are not available.

**Example from Python:**

```python
from __future__ import unicode_l iterals , division , absolute_import
import time
import logging
from collections import deque
try :
from User Dict import Dict Mixin
except ImportError :
from collections import Mapping as Dict Mixin
import six
from six import iteritems
from six . moves import c Pick le
class Base Counter ( object ) :
def __ init __ ( self ) :
raise NotImplementedError
def event ( self , value = 1 ) :
raise NotImplementedError
def value ( self , value ) :
raise NotImplementedError
@ property
def avg ( self ) :
raise NotImplementedError
@ property
def sum ( self ) :
raise NotImplementedError
def empty ( self ) :
raise NotImplementedError
class Total Counter ( Base Counter ) :
def __ init __ ( self ) :
self . cnt = 0
def event ( self , value = 1 ) :
self . cnt += value
def value ( self , value ) :
self . cnt = value
@ property
def avg ( self ) :
return self . cnt
@ property
def sum ( self ) :
return self . cnt
def empty ( self ) :
return self . cnt == 0
class Average Window Counter ( Base Counter ) :
def __ init __ ( self , window _ size = 300 ) :
self . window _ size = window _ size
self . values = deque ( maxlen = window _ size )
def event ( self , value = 1 ) :
self . values . append ( value )
value = event
@ property
def avg ( self ) :
return self . sum / len ( self . values )
```

**Example from Java:**

```
package org . odd job . values ; import java . util . LinkedHashMap
; import java . util . Map ; import org . apache . commons . bean
utils . expression . Default Resolver ; import org . apache . commons
. bean utils . expression . Resolver ; import org . odd job . a ro oa
. A ro oa Exception ; import org . odd job . a ro oa . A ro oa Value ;
import org . odd job . a ro oa . reflect . A ro oa Property Exception
; import org . odd job . a ro oa . reflect . PropertyAccessor ; import
org . odd job . framework . Simple Job ; public class Set Job extends
Simple Job { private final Map < String , A ro oa Value > values = new
LinkedHashMap < String , A ro oa Value > ( ) ; public void setValues (
String name , A ro oa Value value ) { values . put ( name , value ) ; }
protected int execute ( ) throws Exception { for ( Map . Entry < String
, A ro oa Value > entry : values . entrySet ( ) ) { String name = entry
. getKey ( ) ; A ro oa Value value = entry . getValue ( ) ; logger ( ) .
info ( " Setting [" + name + "] = [" + value + "]" ) ; setProperty ( name
, value ) ; } return 0 ; } private void setProperty ( String property , A
ro oa Value value ) throws A ro oa Property Exception { Resolver resolver =
new Default Resolver ( ) ; String comp Name = resolver . next ( property
) ; String property Expression = resolver . remove ( property ) ; if (
property Expression == null ) { throw new A ro oa Exception ( "" ) ; }
Object component = getA ro oa Session ( ) . getBean Registry ( ) . lookup
( comp Name ) ; if ( component == null ) { throw new A ro oa Exception (
"" + comp Name + "]" ) ; } PropertyAccessor property Accessor = getA ro
oa Session ( ) . get Tools ( ) . getProperty Accessor ( ) ; property
Accessor = property Accessor . accessor With Conversions ( getA ro oa
Session ( ) . get Tools ( ) . getA ro oa Converter ( ) ) ; property
Accessor . setProperty ( component , property Expression , value ) ; }
}

 package org . rub ype ople . r dt . internal . ui . util ; import
java . util . Comparator ; import java . util . HashSet ; import java
. util . Set ; import java . util . Vector ; import org . ec lipse
. j face . util . Assert ; import org . ec lipse . j face . view ers
. I Label Provider ; import org . ec lipse . sw t . S WT ; import org
. ec lipse . sw t . events . Dis pose Event ; import org . ec lipse
. sw t . events . Dis pose Listener ; import org . ec lipse . sw t .
events . Selection Listener ; import org . ec lipse . sw t . graphics
. Image ; import org . ec lipse . sw t . layout . Grid Data ; import
org . ec lipse . sw t . layout . Grid Layout ; import org . ec lipse .
sw t . widgets . Composite ; import org . ec lipse . sw t . widgets
. Event ; import org . ec lipse . sw t . widgets . Table ; import
org . ec lipse . sw t . widgets . Table Item ; public class Filter
edList extends Composite { public interface Filter Matcher { void setFilter
( String pattern , boolean ignoreCase , boolean ignore W ild Cards ) ;
boolean match ( Object element ) ; } private class Default Filter Matcher
implements Filter Matcher { private String Matcher f Matcher ; public void
setFilter ( String pattern , boolean ignoreCase , boolean ignore W ild
Cards ) { f Matcher = new String Matcher ( pattern + '*' , ignoreCase ,
ignore W ild Cards ) ; } public boolean match ( Object element ) { return f
Matcher . match ( f Renderer . getText ( element ) ) ; } }
```

**Example from Go:**

```go
func Average Color ( colors ...  Color ) ( color Color ) { var ( x , y
float 64
hue , sat , b ri , kel int
)

// Sum s ind / cos d for h ues for _ , c := range colors { // Convert hue
to degrees h := float 64 ( c .  H ue ) / float 64 ( math .  Max Uint 16 ) *
360 .  0

x += math .  C os ( h / 180 .  0 * math .  Pi )
y += math .  S in ( h / 180 .  0 * math .  Pi )
sat += int ( c .  Sat uration )
b ri += int ( c .  B right ness )
kel += int ( c .  K el vin )
}

// Average s ind / cos d x /= float 64 ( len ( colors ) )
y /= float 64 ( len ( colors ) )

// Take atan 2 of aver aged hue and convert to uint 16 scale hue = int (
( math .  At an 2 ( y , x ) * 180 .  0 / math .  Pi ) / 360 .  0 * float 64
( math .  Max Uint 16 ) )
sat /= len ( colors )
b ri /= len ( colors )
kel /= len ( colors )

color .  H ue = uint 16 ( hue )
color .  Sat uration = uint 16 ( sat )
color .  B right ness = uint 16 ( b ri )
color .  K el vin = uint 16 ( kel )

 return color
}
func Color Equal ( a , b Color ) bool { return a .  H ue == b .  H ue &&
a .  Sat uration == b .  Sat uration && a .  B right ness == b .  B right
ness && a .  K el vin == b .  K el vin
}
func ( s * Subscription ) notify ( event interface { } ) error { timeout :=
time .  After ( Default Timeout )
select { case <- s .  quit Chan :  Log .  Debugf ( " " , s .  id )
return Err Closed
case s .  events <- event :  return nil
case <- timeout :  Log .  Debugf ( " " , s .  id )
return Err Timeout
}
}
```

