# OpenReview forum: "NoEsis: A Modular LLM with Differentially Private Knowledge Transfer"
_ICLR.cc/2025/Workshop/MCDC — MCDC @ ICLR 2025_

### Official Review · Reviewer_cvim · 2025-02-28

**Rating:** 7
**Confidence:** 4
**Fit:** 5

**Summary:**

The paper proposes NOESIS, a novel framework that integrates differential privacy into a modular adaptation of large language models. The approach is based on a two-stage, parameter-efficient fine-tuning method. In the first stage, a shared set of trainable prompt tokens is trained under differential privacy, enabling safe knowledge transfer across domains. In the second stage, domain-specific low-rank adapters (Mix-LoRA) are tuned on individual private datasets/domains. The method is evaluated on a code completion task across three programming languages (Python, Java, and Go) and shows that NOESIS bridges a substantial part of the accuracy gap between non-private and non-shared models while reducing privacy leakage, as evidenced by empirical membership inference attacks.

**Reason For Giving A Higher Score:**

- The problem is relevant and the solution is innovative.
- Overall the empirical methodology is robust and rigorous.
- The results on the particular task are relevant, outperforming previous solutions.

**Reason For Giving A Lower Score:**

- The narrow evaluation focus (only code completion) limits the impact.
- Baseline choices lacking a comparison to a strong baseline (DP-trained MixLoRA).
- No clear discussion of negative transfer effects, which could impact performance in certain domains.

**Strengths And Weaknesses:**

Strengths
1. The proposed multi-step approach is novel. Despite existing work already exploring differential privacy in parameter-efficient fine-tuning or modular learning separately, the combination of Mix-LoRA (modular PEFT) with differentially private prompt tuning is innovative, and NOESIS merges these approaches effectively.
2. The work addresses a critical issue in AI, privacy-preserving multi-domain learning, which is highly relevant given rising concerns over LLM data leakage and different regulations over the globe.
3. The work has a clear and rigorous empirical methodology, with hypothesis clear and extensive ablation studies on the effect of each component. The method and experimental setting choices are well explained.
4. The method also presents strong results, bridging 77% of the accuracy gap between non-private and fully private models, and outperforming DP and PEFT baselines. Also, experiments demonstrate robust knowledge transfer to low-resource language (Go).

Weakness
1. The experiments on knowledge transfer do not fully substantiate the paper’s claims. The authors should have included baselines that explicitly demonstrate the benefit of having shared parameters across all languages, including the high-resources. For instance, what if training on Java+Python alone outperforms Java+Python+Go? Similarly, could Noesis trained only on Python achieve better results than the full multi-domain settings? Analyzing potential cross-domain transfer degradation would strengthen the claims regarding knowledge sharing.
2. A key missing baseline is Mix-LoRA with DP-SGD training (on out-of-domain data). This would provide a direct comparison to assess the effectiveness of differentially private prompt tuning and determine whether shared prompts are an efficient contribution to knowledge transfer beyond domain-specific fine-tuning, or if its effects can be reached with a simpler approach.
3. The experiments focus on one specific task (code completion) which not necessarily would generalize to all cases (but this is acceptable considering the scope of the workshop).

**Suggestions:**

- Could the authors clarify the distinction between "NOESIS with Common LoRA" as shown in Figure 4 and the "Single Common Adapter"? Are these two names referring to the same model, or does "NOESIS with Common LoRA" denote the variant mentioned in line 252 that employs a LoRA as a common knowledge-sharing backbone?
- The paper would benefit from a more detailed discussion on how the domain-specific experts are merged during deployment. For example, If I want to use only Python and Java languages in production, are the experts merged? And how does this affect performance?
- Can the authors elaborate on the functioning of the domain router? Specifically, is the routing learned and performed dynamically, as in the original MixLoRA? Or is it handled manually by loading and unloading the relevant LoRA experts when switching datasets?
- There appears to be a notation issue in Equations 1 and 2, what does the W mean?
- It would strengthen the claims if the authors considered a stronger baseline where a domain-specific LoRA are also trained on other domains using DP-SGD. Do the authors think such a model might outperform NOESIS? A comparative analysis here could provide valuable insights into the advantages of the proposed approach.
- The "Experiments Results" subsection could be better structured for clarity. However, I understand the lack of space given by the workshop constraints.

---

### Official Review · Reviewer_3NH4 · 2025-03-04

**Rating:** 9
**Confidence:** 4
**Fit:** 5

**Summary:**

The authors propose a modular framework that aims to enable both privacy between domains but also knowledge transfer.

They note that private learning is leading to poor performance, and MoE to leakage of information. Thus, they propose NoEsis:
* first, tune prompt tokens, using differentially private learning mechanisms, allowing transfer between tasks/domains
* second, train a MixLoRA, with different adapters per tasks/domains

The domains considered, for a LM, are the Python, Java, and Go programming languages, using a 220M decoder model.
The two main baselines are 1) sharing nothing -- a lower bound, and 2) sharing everything and thus leaking -- a potential upper bound.

**Reason For Giving A Higher Score:**

The problem tackled makes sense, the performance (particularly table 2) are good, and the methodology is sound.

Moreover, it fits nicely the topic of the workshop, about exploring new approaches to modularity, here privacy-oriented modularity, which is seldom explored to my knowledge.

**Reason For Giving A Lower Score:**

I'm not convinced about table 3. I'm not sure how well the results hold at a larger scale than 220M and more diverse tasks than coding. I'd have liked more discussions on some details (see my suggestions).

However, it's an excellent workshop paper in my opinion.

**Strengths And Weaknesses:**

# Strengths

In no particular order.

1. NoEsis tackles efficiently two directions, which are usually opposed:
* Enabling privacy-oriented methods is critical for taping more data and personalization of LM
* Mixture-of-Experts reach best performance, but leakage of information between domains experts is a problem

2. Main results on table 2 are impressive, where the proposed method reaches better performance than per-domain specialist LM while avoiding unwanted leakage between said domains

3. Method is simple, based principally on two well studied methods (prompt tuning + mixture of LoRAs), and would benefit seamlessly from improvements of any of those two methods

4. Code is provided

# Weaknesses

In no particular order.

1. Table 3 evaluates varying rank for the common LoRa and varying prompt size. However, the performance itself doesn't seem to vary much, how much of it is significant?

2. Would the results hold on other tasks than coding?

**Suggestions:**

* instead of private domains, how would you extend to private tokens? e.g. Python knowledge can leak to the Java knowledge, but some specific tokens (let's say user-defined ids) shouldn't be leaked: aka changing the granularity of the privacy
* can we streamline the procedure in a single unified stage for simplicity, training at the same time the prompt and lora?
* in the deployment phase, what's the performance if all LoRAs are merged together?
* what's the performance of DP prompt tuning, without MixLoRA?

---

### Decision · Program_Chairs · 2025-03-06

**Decision:**

Accept

**Comment:**

This work proposes to bring private learning to modular systems through a two-stage procedure, reaching good performance while avoiding leaking information from one domain to another. Learning efficiently across several domains without leaking info is relevant for collaborative learning and this work. The reviewers all recommend acceptance, and we're happy to accept it to this workshop.